# The Effect of Acetylation on Iron Uptake and Diffusion in Water Saturated Wood Cell Walls and Implications for Decay

**Samuel L Zelinka** [1], **Carl J. Houtman** [1], **Kolby Hirth** [1], **Steven Lacher** [1], **Linda Lorenz** [1], **Emil Engelund Thybring** [2] **and Christopher G. Hunt** [1,*]

[1]  USDA Forest Service, Forest Products Laboratory, Madison, WI 53726, USA;
samuel.l.zelinka@usda.gov (S.L.Z.); carl.houtman@usda.gov (C.J.H.); kolby.c.hirth@usda.gov (K.H.);
steven.j.lacher@usda.gov (S.L.); Linda.f.lorenz@usda.gov (L.L.)

[2]  Department of Geosciences and Natural Resource Management, University of Copenhagen,
DK-1958 Frederiksberg C, Denmark; eet@ign.ku.dk

*  Correspondence: christopher.g.hunt@usda.gov; Tel.: +1-608-231-9521

**Abstract:** Acetylation is widely used as a wood modification process that protects wood from fungal decay. The mechanisms by which acetylation protects wood are not fully understood. With these experiments, we expand upon the literature and test whether previously observed differences in iron uptake by wood were a result of decreased iron binding capacity or slower diffusion. We measured the concentration of iron in 2 mm thick wood sections at 0, 10, and 20% acetylation as a function of time after exposure to iron solutions. The iron was introduced either strongly chelated with oxalate or weakly chelated with acetate. The concentrations of iron and oxalate in solution were chosen to be similar to those found during brown rot decay, while the concentration of iron and acetate matched previous work. The iron content of oxalate-exposed wood increased only slightly and was complete within an hour, suggesting little absorption and fast diffusion, or only slight surface adsorption. The increase in iron concentration from acetate solutions with time was consistent with Fickian diffusion, with a diffusion coefficient on the order of $10^{-16}$ m$^2$ s$^{-1}$. The rather slow diffusion rate was likely due to significant binding of iron within the wood cell wall. The diffusion coefficient did not depend on the acetylation level; however, the capacity for iron absorption from acetate solution was greatly reduced in the acetylated wood, likely due to the loss of OH groups. We explored several hypotheses that might explain why the diffusion rate appears to be independent of the acetylation level and found none of them convincing. Implications for brown rot decay mechanisms and future research are discussed.

**Keywords:** acetylation; wood modification; diffusion; brown rot decay; iron

## 1. Introduction

Brown rot decay fungi are a class of basidiomycetes known to cause rapid strength loss in softwoods, the primary type of wood used in construction [1,2]. Case studies have shown that over 85% of wood decay failures are caused by brown rot fungi [3]. The first stages of brown rot decay are believed to involve the diffusion of small, chelator-mediated Fenton (CMF) reagents into the wood cell wall, which causes radical depolymerization of the biomass [4,5]. CMF could also be supported by the observed fungal translocation of iron into wood during early decay [6–8]. The CMF reactions are consistent with observations that the strength of wood exposed to brown rot decreases before wood mass loss occurs [9–11].

High levels of wood acetylation have been shown to inhibit brown rot decay [9,12–14]. In contrast to preservative treated wood, which protects the wood through fungi-static or fungi-toxic methods, acetylation alters the cell wall polymers in a non-toxic manner to make them a less suitable substrate for decay fungi [15]. Acetylation has been studied since 1928 [16], and involves the replacement of OH hydrogens in the cell wall with acetyl groups. The level of acetylation is typically referred to by the weight percent gain (WPG) and is calculated by the change in mass caused by the acetylation process, given in Equation (1)

$$WPG = \frac{m_{post} - m_{pre}}{m_{pre}} \tag{1}$$

where $m_{pre}$ and $m_{post}$ are the dry masses of wood before and after the acetylation process, respectively. It has been shown that above 18–20% WPG, acetylated wood is decay resistant [9,12–14,17,18].

The method by which acetylation provides wood protection is not fully understood and is an area of active research [19,20]. It has recently been proposed that acetylation inhibits brown-rot decay by preventing the diffusion of CMF reagents through the cell wall [19]. The proposed mechanism assumes that diffusion in acetylated wood is much slower than in untreated wood. As a result, the CMF reagents diffuse into and degrade the wood extremely slowly, depriving the fungus of its food source, and therefore preventing structural damage. This proposed mechanism is supported by recent advances in our understanding of diffusion of ions in the wood cell wall. Ionic diffusion through wood cell walls requires large scale molecular motions of amorphous polysaccharides such as hemicellulose, which are strongly affected by cell wall moisture content [21,22]. The onset of these motions can be observed by a mechanical softening known as the glass transition. Observed moisture-induced transitions in wood cell walls were recently reviewed [22]. Further work [23] correlated the frequency of these motions with the rate of ion mobility measured by Zelinka [24–26]. Since acetylation changes both the equilibrium moisture content of wood and the cell wall polymers themselves [15], it is plausible that acetylation could affect hemicellulose softening and therefore diffusion. Indeed, XFM measurements on acetylated wood in the hygroscopic regime show that acetylation increases the relative humidity threshold for ion diffusion in acetylated wood [27].

While these developments show that there may be a link between acetylation, diffusion, and decay resistance, the measurements have been conducted in wood conditions at air relative humidity (RH) levels less than 95% (i.e., the hygroscopic region). While decay can be observed in moisture contents as low as 20% [28,29], decay fungi are most active when the water potential is above −4 MPa, or 97% RH [30]. In this moisture regime, the wood cell walls should be nearly saturated and liquid-like water should exist in the cell lumina [31–38]. Therefore, the previous experiments conducted below 95% RH cannot fully test whether acetylation inhibits decay by preventing diffusion when the cell walls are nearly or fully saturated.

The only experiments to examine iron uptake in acetylated wood at saturation were performed by Hosseinpourpia and Mai [39], who examined the uptake of a dilute solution of iron (II) acetate into thin veneers after 48 h. The amount of iron in the wood was determined using inductively-coupled-plasma (ICP) spectroscopy of the digested wood. The results indicated that untreated wood absorbed 16 times more iron than the highly acetylated wood. While those data suggest that acetylation may inhibit diffusion under saturated conditions, they are not conclusive. Since only one time point was observed, it is impossible to determine whether the acetylated wood had less iron because iron diffused more slowly into the acetylated wood or rather because acetylated wood had a lower binding capacity for iron. The single time point does not indicate whether the observed iron content reached the saturation level. In this study, we build upon the previous work of Hosseinpourpia and Mai [39] by using ICP to examine how acetylation affects iron uptake in wood cell walls at several different time points with two different buffer solutions.

## 2. Materials and Methods

Loblolly pine (*Pinus taeda* L.) sapwood samples were cut 2 mm thick in the longitudinal direction to maximize transport through the lumina. All samples were cut from the same board and free from compression wood. Samples were acetylated using a method similar to our previous work fully described by Passarini et al. [40] Briefly, the samples were placed in a bath of acetic anhydride, vacuum infiltrated, heated to 140 °C, and removed at various time points to achieve the desired weight percent gains. In addition to untreated wood, two levels of acetylation were examined, 10% WPG (9.6–10.8%) and 20% WPG (19.6–21.5%).

The uptake experiments were started by making a series of nylon mesh ribbons with ten pockets, one for each replicate specimen. The specimens were placed in their pockets (Figure 1) and soaked in an iron-free 1 mM Na/H oxalate buffer solution (pH 4.0) at room temperature. Several cycles of immersed vacuum, shaking, and vacuum release, followed by at least an hour of soaking, were used to ensure that the wood was fully saturated and to separate out the effects of water absorption from iron diffusion. Solution iron and oxalate concentrations were chosen to mimic what is found in lumen water of wood experiencing brown rot decay [41–43] (see Supplementary Materials for a full discussion). A gently stirred, 200 L reservoir of identical buffer with an added 100 μM iron (III) (actual 109 μM, as $FeCl_3$) was also prepared, large enough that potential iron uptake by samples would not affect the solution concentration. Assuming these conditions result in an equilibrium among the possible iron chelation states [44,45], one would expect that 83% of the iron ions will be complexed with three oxalates, with the balance associated with two (Figure S1: iron species concentration for a solution of 100 μM iron(III) chloride in 1000 μM oxalate buffer at various pH values) [46]. Sampling of the iron oxalate solution at the end of the experiment confirmed the bulk solution concentration did not change. After wetting in iron-free buffer, the wood sections were placed in the reservoir of iron-containing solution for specified times. After removal from the iron solution and nylon ribbon, excess water was removed from samples by placing between sheets of absorbent paper and rolling a 13 kg cylinder over the specimens, and samples were immediately weighed. Subsequently the oven dry weight was obtained after drying at 105 °C for 12+ h. The iron content of each specimen was measured by ICP-OES after digesting the oven-dried samples in nitric acid, similar to the protocol presented in AWPA standard A21 [47], averaging triplicate readings with typical RSDs (standard deviation/average) under 1%.

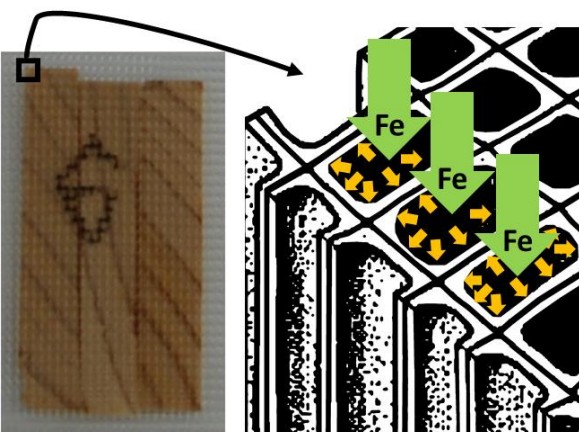

**Figure 1.** Left—1 cm wide specimens in nylon mesh. Right—magnified view schematic of iron complexes diffusing quickly through lumen water (large green arrows) and subsequently into cell walls, the focus of this study (small orange arrows).

The iron present in lumen water was subtracted from the measured iron concentrations so that our values represent the amount of iron in the cell walls. The mass of lumen water was calculated as (total mass–oven dry mass–bound water mass). The bound water content in the saturated cell walls

was determined in a separate experiment using solute exclusion. This cell wall "bound water capacity" (formerly called fiber saturation point or FSP, but that term is problematic [30,32]) was found to be 42% (mass, dry basis) for the control wood, 30.5% for the 10% WPG samples, and 18% for the 20% WPG samples; similar to published values using the same method [48]. The concentration of iron in the cell lumina was assumed to be equal to that of the bulk solution. Iron concentrations are given per gram of oven dry wood or modified wood.

The experiments were repeated with a 200 µM $FeCl_3$, 100 mM acetate pH 4.0 solution (actual 213 µM $FeCl_3$), made with acetic acid, sodium acetate, and NaOH, after wetting new samples in iron-free acetate buffer. The goal of these experiments was to directly compare our findings to those of Hosseinpourpia and Mai [39] by using the same solution concentrations and analysis method, but at multiple time points. All experimental details were the same as with oxalate buffer except for the following. A 200 L master batch of buffer solution was made, and 18 L transferred to each of several polyethylene buckets gently stirred with large stir bars. One mesh ribbon, representing one time point, was incubated in each bucket. After sample removal, the buffer was sampled to verify that iron was not depleted from solution, and fresh solution was used to repeat the process for another time point.

All statistical analyses were conducted using standard Student's t-testing or F-testing implemented in Microsoft Excel spreadsheets. F-testing to obtain confidence intervals for fitted parameters was done following the method of Box et al. [49]

To examine whether acetylation affects the diffusivity, we modeled the uptake of iron. If one can approximate the transport into the wood cell wall as a thin one-dimensional sheet with the same solution concentration on both sides (i.e., adjacent lumina), an equation developed by Crank [50] can be used to predict the amount of iron that has entered the wall as a function of time. In Equation (2), $E(t)$ is defined as the dimensionless fractional change in concentration, which accounts for the significant change in absorption capacity, as seen in Figure 3.

$$E(t) = \frac{c - c_o}{c_\infty - c_o} = 1 - \frac{8}{\pi^2} \sum_{n=0}^{\infty} \frac{1}{(2n+1)^2} e^{-\frac{(\pi(2n+1))^2}{4}\left(\frac{Dt}{L^2}\right)} \tag{2}$$

where $c$ is the concentration at a given time point, $c_o$ and $c_\infty$ are the concentrations at the beginning of the experiment (2.2 µg g$^{-1}$) and at equilibrium, respectively, $D$ (m$^2$ s$^{-1}$) is the diffusion constant, $t$ is time (s), and $L$ is wall thickness (m). The summation shown in Equation (2) was implemented in an Excel macro. The summation was continued until the next term dropped below $10^{-12}$, which typically required less than 15 terms.

For materials that exhibit Fickian diffusion, $E(t)$ should initially be linearly proportional to the square root of time and reach an asymptotic value of 1 at long times. Equation (2) is a rapidly converging series of exponentials with the primary independent variable being dimensionless time shown as Equation (3).

$$\tau = \frac{D\,t}{L^2} \tag{3}$$

## 3. Results

The results of soaking wood in iron oxalate solutions are shown in Figure 2. A significant difference can be seen between the measurements on wood soaked without Fe, which had an average of 2.2 µg g$^{-1}$ and those measurements after soaking in the Fe solution, which had an average of 4.8 µg g$^{-1}$ (µg Fe per g dry wood), an increase of 2.6 µg Fe per g of wood. For perspective, while saturated unmodified wood contains 0.42 g of water per g of wood, 0.42 g of this buffer contains 2.5 µg of Fe. For purposes of statistical analysis, all data for non-zero times were collected as a single pooled set, and then the average for each time and condition point was t-tested against this pooled set. Results with a $p < 0.05$ are indicated on Figure 2 with an asterisk. Within the accuracy of this experiment and this statistical test, it appears that iron uptake is nearly complete within one hour. The final time

point data may indicate that there is a slow increase in iron concentration over time, but the data are insufficient to make a strong conclusion.

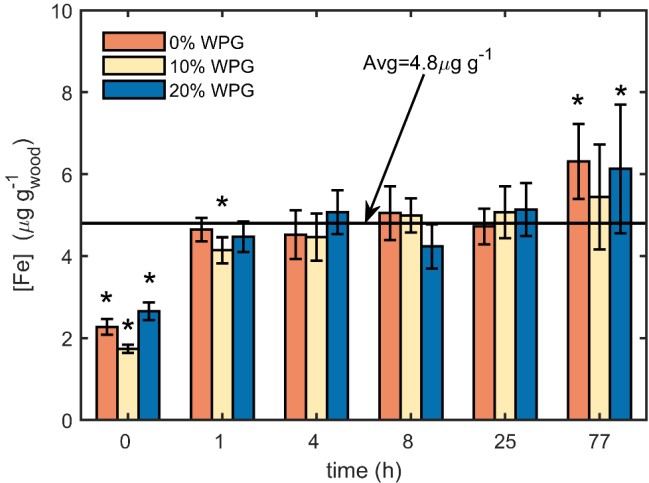

**Figure 2.** Concentration of iron in wood sections soaked in a 100 μM iron + 1 mM oxalate solution as a function of the level of acetylation and time in solution. Error bars represent the 95% confidence interval. * indicates that a data set is significantly different ($p < 0.05$) from the grand average of 4.8 μg g$^{-1}$.

The hypothesis that there was no difference among the three levels of acetylation was also subjected to *t*-testing. For each time point greater than zero, the means of three data sets were pairwise compared to each other, which results in three independent tests. Only one of the time points, 8 h, had one average that was significantly different from the other two ($p < 0.05$), 20% WPG versus 0% and 10% WPG. Again, within the limits of the data, there does not appear to be a significant difference between the three acetylation levels.

In contrast to the very small amount of iron uptake found with the oxalate solution, large amounts of iron were absorbed from the iron acetate solution, and the amount of iron depended upon both the exposure time and the level of acetylation, as shown in Figure 3. Note the 150× difference in the vertical scales of Figures 1 and 2. For the unacetylated wood, the amount of iron absorbed from the acetate solution was 250 times greater than that from the oxalate solution.

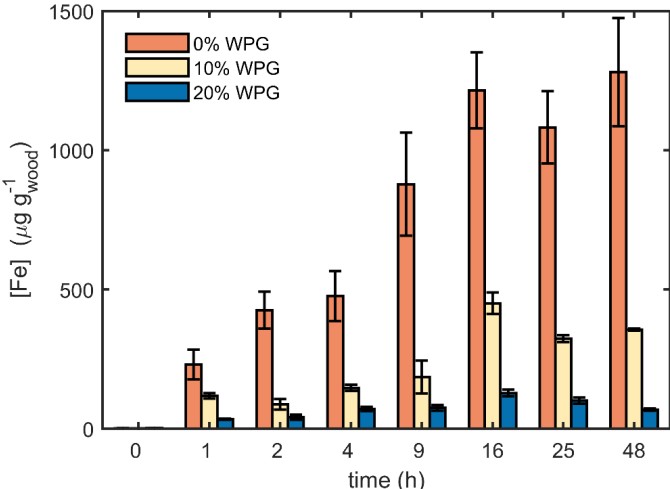

**Figure 3.** Concentration of iron in wood sections soaked in a 200 μM iron + 100 mM acetate solution as a function of the level of acetylation and time in solution. Error bars represent the 95% confidence interval.

Inspection of Figure 3 shows that acetylation affects the capacity for iron absorption in the cell walls. By absorption we mean located with the bound water of the cell wall as opposed to adsorption on surfaces in contact with free water. For the 16 h data, the 10% and 20% WPG wood absorbed 37% and 10% as much iron, respectively, as control wood, following the same trend observed by Hosseinpourpia and Mai [39].

Using Equation (2), the uptake data for all three levels of WPG from the acetate solution have been replotted as a fraction of final Fe concentration in Figure 4. Also shown in Figure 4 is a curve representing the best fit of the model as determined by least squares. Four parameters were used to fit the data: one $c_\infty$ for each of the three data sets and a single value for characteristic time, $L^2/D$. A nested F-test [49] was used to determine if there was justification for using a separate value for $L^2/D$ for each of the data sets. The results of this test gave a $p = 0.56$, so the data do not show evidence for a difference in $L^2/D$ for the three levels of acetylation. Diffusion from the bulk solution to the specimen center via lumen should take approximately 15 min (small molecules in water $D \approx 1 \times 10^{-9}$ m$^2$ s$^{-1}$), suggesting that iron depletion in the lumen is not likely affecting these results.

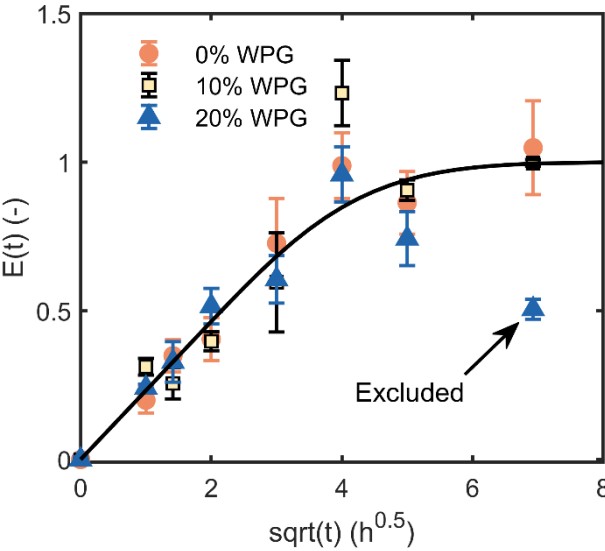

**Figure 4.** Iron(III) uptake from acetate solution plotted as $E(t)$, or the fractional change in concentration, as a function of the square root of time along with the model fit.

The uptake data for all three data sets do appear to exhibit an initial linear dependence on the square root of time and arrive at an asymptote. The final time point of the 20% WPG appears to be an outlier and was excluded from the analysis. It is rather unlikely that the uptake would drop so significantly at a long time.

Because $D$ and $L$ are lumped in the model as $D/L^2$, we can estimate $D$ only if we have knowledge of the distance from the center of the cell wall to the lumen surface ($L$). These samples have a bimodal distribution of cell wall thicknesses for earlywood (~5 μm lumen to middle lamella) and latewood (~8 μm) cell walls. Since the main purpose of this work was to examine differences caused by acetylation, a single value of 6 μm was used to simplify calculations and compare treatments. While this affects the absolute value of the diffusion coefficients, the comparisons between treatments should still be valid.

The best fit values for $c_\infty$ were 1230, 353, and 127 μg g$^{-1}$ for 0%, 10%, and 20% WPG, respectively. The best fit value for $D/L^2$ was $1.2 \pm 0.7 \times 10^{-5}$ s$^{-1}$. Assuming a value for $L$ of 6 μm, the corresponding 95% confidence interval for $D$ is 2 to $7 \times 10^{-16}$ m$^2$ s$^{-1}$. These diffusion coefficients are significantly lower than what may be expected for other ions in wood, which are often several orders of magnitude higher, $10^{-15}$–$10^{-10}$ m$^2$ s$^{-1}$ [51]. Thus, the unusually tight binding of Fe(III) to wood [52–55], is clearly slowing the diffusion. While there is scatter in the data and the diffusion lengths are ill-defined, it is clear that the diffusion of iron in the acetylated wood is not orders of magnitude lower than untreated wood as

predicted if the mechanism by which acetylation inhibits brown rot decay is hindering diffusion [19]. In fact, the diffusion of iron(III) in acetylated wood under saturated conditions in acetate buffer appears to occur at approximately the same rate as in untreated wood.

## 4. Discussion

The data show clear differences in the uptake rate and absorption capacity of iron from the two different buffer solutions. Given the very small uptake, diffusion coefficients for iron(III) in the wood cell wall from oxalate solution could not be obtained from the data. The slight uptake at one hour could be explained by three hypotheses: (1) the samples reached equilibrium before our first time point at 1 h, (2) surface adsorption, such as on cell wall lumina and pit membranes dominated on the time scale of this experiment, or (3) there is a systematic under-correction for the iron in the lumen water. While surface adsorption (hypothesis 2—surface adsorption) is very fast, we find it strange that adsorption would be the same on acetylated and unmodified wood surfaces. Hypothesis 3 is unlikely since we have thoroughly reviewed our calculations. Therefore we focus on hypothesis 1: diffusion is largely complete in one hour.

Calculations of solution equilibria show that most of the iron in our oxalate solution will be associated with three oxalate ions (Figure S2). These complexes will have a net −3 charge that will not interact strongly with the wood matrix, which is also anionic. Oxalate has been previously shown to bind iron more effectively than wood [52]. Jakes et al. [51] recently used X-ray fluorescence microscopy to measure the diffusion coefficients of K+ and Cu(II) in the hygroscopic region in wood cell walls. Using chloride as a counter-ion, they found that at 75% RH, the diffusion coefficient of Cu(II), was approximately $3 \times 10^{-14}$ m$^2$ s$^{-1}$ and that of K$^+$ was approximately $3 \times 10^{-13}$ m$^2$ s$^{-1}$. Assuming $D$ for iron-oxalate complexes to be between $10^{-14}$ and $10^{-13}$ m$^2$ s$^{-1}$ and an $L$ of 6 μm, the characteristic time, $L^2/D$ is expected to be between 6 and 60 min, indicating that the 0% WPG wood, at least, should be approaching saturation within the first hour. However we expect that the radius of the ion, accompanying counterions, and hydration spheres of the dominant ion in this experiment, Fe(oxalate)$_3$(H$_2$O)$_3$ $^{-3}$, is much larger than the ions measured by Jakes et al., which would be expected to significantly slow diffusion through the cell wall polymer matrix. If diffusion were dramatically hindered, these experiments might be too short to observe the diffusion. Moreover, we expect acetylation to slow diffusion as it reduces the void volume for diffusion and the water content in the saturated cell wall [48]. It seems odd that diffusion is also complete in the acetylated samples after only one hour. Therefore, from these data it is not possible to distinguish the relevant mechanism driving the uptake during the first hour. The low level of Fe(III) uptake from oxalate solution overall, however, is likely a result of the high formation constants for oxalate complexes (see appendix), as well as the exclusion of negative Fe(oxalate) ions from the negatively charged wood matrix by the Donnan effect [56,57].

Interestingly, the current understanding of brown rot decay involves the release of iron and oxalate from the fungal hyphae [5]. Since the wood cell walls of decayed wood are enriched with iron [6,8], the simplest explanation is that these iron oxalates are freely diffusing into the cell wall. However, the data in Figure 2 and the observed suppression of oxidation-reduction cycling of iron by high oxalate and low pH conditions [58,59] highlights the complexity of chelator-mediated Fenton chemistry in the early stages of brown rot decay.

The experiments performed with iron acetate solutions with unchelated iron (See Supplementary Material), showed a strong association between iron and wood at all levels of acetylation and a clear reduction in iron absorption capacity after acetylation, with a 90% reduction at 20% WPG. High association constants for iron with biomass are widely known. For example, biomass has been considered as a medium to immobilize iron for arsenic remediation [60]. Arantes et al. [52] showed that fixed beds of milled wood could completely remove iron from dilute solutions. This strong association is expected to reduce the effective diffusion coefficient, $D_{eff}$. The case of diffusion when the diffusant associates with the immobile network has been treated by Crank [50]. Equation (4) is an

expression for an effective diffusion coefficient as a function of the normal diffusion coefficient and the chromatographic retention factor, $K$.

$$D_{\text{eff}} = \frac{D}{1 + K} \tag{4}$$

Our estimate of the iron diffusion coefficient in acetate buffer, on the order of $10^{-16}$ m$^2$ s$^{-1}$ with no apparent dependence on the amount of acetylation, was for $D_{\text{eff}}$. As noted above, Jakes et al. [51] estimated ion diffusion coefficients for other ions between $10^{-14}$ and $10^{-13}$ m$^2$ s$^{-1}$. Furthermore, electrical conductivity in wood [25,61] and the results of Jakes et al. [51] suggest that the diffusion coefficient should increase with wood moisture content. Under saturated conditions, the diffusion coefficients of Cu ions are estimated to be between $1 \times 10^{-15}$ to $9 \times 10^{-12}$ m$^2$ s$^{-1}$ [51,62,63]. If the value of the retention factor for iron(III) ions with wood is $K > 100$, then the observed $D_{\text{eff}}$ near $10^{-16}$ m$^2$ s$^{-1}$ could be expected. The chromatography literature using a wood stationary phase [52–55] suggests that a $K$ of this magnitude for Fe(III) is not unreasonable.

The observation that the diffusion coefficient did not depend on the level of acetylation was unexpected. We expect acetylation to slow diffusion as it reduces the void volume for diffusion and the water content in the saturated cell wall [48]. We discuss two possible explanations. Rowell et al. [64] have previously shown that acetylation of thoroughly dried wood with dried acetic anhydride sometimes does not proceed uniformly. Our experiments may be only measuring transport and absorption in unacetylated regions of the wood, and little transport occurs in the acetylated regions on the time scale of these experiments. This is unsatisfying because we would expect the slow diffusion acetylated regions to be on the surface, effectively blocking access to the faster diffusing regions. Another possible hypothesis is that acetylated wood has two competing effects: lower polymer relaxation rates slowing diffusion, and a lower chromatographic retention factor $K$ because of the loss of OH groups, increasing diffusion. We find it unlikely that these two effects would exactly balance each other out. An alternative hypothesis is that the observed iron is simply adsorbed on the wood surfaces. Fe(III) is known to form orange precipitate in acetate solutions, which could accumulate on any surface, including the wood. This hypothesis has two problems; first, there was no hint of orange in the solution even at the 48 h time point when viewing the bottom of a white bucket through 20 cm of solution. Second, it is unclear why the accumulation of Fe on wood surfaces would stop after 16 h. Clearly we lack sufficient evidence to explain the lack of a difference between diffusion rates at different acetylation levels.

There are many small molecules that could support CMF reactions inside the cell wall even while iron is immobile [65]. Therefore experiments with a diffusant with less complicated behavior than iron might still provide useful insights into the role of diffusion in decay resistance of acetylated wood. Moreover, as iron mobility is not needed for CMF reactions, the lower Fe binding capacity of acetylated wood could be a protection mechanism, if the lower iron content inside the cell wall results in a lower CMF reaction rate. The presence of oxalate, however, seems to erase this potential advantage of acetylated wood.

Our methodology of using ICP to examine the uptake of an inorganic diffusant is a viable method for measuring diffusion coefficients and binding capacities under water saturated conditions, but more work is needed to better understand how these experiments relate to the degradation of wood by fungally produced CMF reagents and how acetylation affects this degradation. A full understanding of the role acetylation plays in inhibiting CMF degradation requires a better understanding of the relevant diffusants and their solubility, adsorption, binding capacity, redox potentials, and diffusivity in native and modified wood cell walls.

## 5. Conclusions

This study examined the in vitro diffusion of iron oxalate and iron acetate into fully water saturated wood cell walls, at concentrations relevant to the brown rot decay process. It was found that only a small amount of iron oxalate was taken up by the wood cell wall. In contrast, the exposure to iron

acetate solution resulted in a dramatic, time-dependent increase in the iron concentration in the wood cell wall.

The absorption of the iron from acetate solution into the cell wall was consistent with Fickian diffusion. By utilizing diffusion models based upon the fractional change in concentration, $E(t)$, it was possible to estimate the diffusion coefficient of the iron under water-saturated conditions. This analysis was complicated by uncertainties in the maximum asymptotic concentration and the range of potential cell wall thicknesses. Given these uncertainties, the diffusion coefficients ($D_{eff}$) were estimated to be between $2 \times 10^{-18}$ and $7 \times 10^{-16}$ m$^2$ s$^{-1}$. Diffusion rates did not appear to depend on the level of acetylation; this suggests that much remains to be understood about the relationship between acetylation, diffusion, and brown rot decay.

The data suggest that for the iron + acetate system examined, the diffusion rate is not affected by acetylation, and that for the iron + oxalate system, the acetylated wood did not appear to significantly hinder diffusion. While a full understanding of the effect of acetylation on CMF degradation cannot be discerned from these data, acetylation may affect CMF degradation by lowering the capacity for iron or by other means than iron diffusion.

**Supplementary Materials:** The following are available online at http://www.mdpi.com/1999-4907/11/10/1121/s1, Figure S1: iron_oxalate, Figure S2: iron_acetate.

**Author Contributions:** Conceptualization, S.L.Z., E.E.T., C.G.H.; methodology, all; measurements, K.H., S.L., L.L., C.G.H.; formal analysis and writing, S.L.Z., C.J.H., C.G.H.; reviewing and editing: all; project administration, C.G.H.; funding acquisition C.G.H. All authors have read and agreed to the published version of the manuscript.

**Funding:** The authors salaries were funded by their respective institutions. DOE BER grant DE-SC0012742 funded the solute exclusion experiments through salary for Hideki Suzuki.

**Conflicts of Interest:** The authors declare no conflict of interest.

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
