# Peer review of "The Effect of Acetylation on Iron Uptake and Diffusion in Water Saturated Wood Cell Walls and Implications for Decay"

_forests, doi:10.3390/f11101121_

Round 1
Reviewer 1 Report
Comments related to the manuscript by Samuel L. Zelinka, Carl J. Houtman, Kolby Hirth, Steve Lacher, Linda Lorenz, Emil Engelund Thybring and Christopher G. Hunt “The effect of acetylation on iron uptake and diffusion in water saturated wood cell walls and implications for decay” submitted for publishing in Forests:
The article is an attempt to determine whether the acetylated wood had less iron because iron diffused more slowly into the acetylated wood or rather because acetylated wood had a lower binding capacity for iron. The explanation of this issue is extremely important to understand the phenomena of resistance of wood against brown decay. I am convinced that the paper delivers useful knowledge in the area of wood modification.
The way of formulating research hypotheses should be considered correct. The manuscript was written clearly and logically. However, some minor remarks should be considered by the Authors:
Page 1 line 6.
Please place the affiliation number “2” after the author name: Emil Englund Thrybing.
Page 2 line 71.
I suggest replace the term "relative humidity” with "air relative humidity". The last term is more appropriate in psychorometrics.
Page 2 line 74.
Please move the acronym explanation (air RH) to line 71.
Page 2 line 72.
I suggest replace the term “the hygroscopic region” with “the bound water content”.
Page 2 line 72.
Please consider including the other specimen dimensions used in the experiments.
The content of the previous work (Passarini et al 2017), shows that the dimensions of the specimen in the radial direction were from 2 to 20 mm.
Page 3 line 116.
Please replace the term "overnight" with a more specific term, e.g. drying time in hours.
Page 3 line 122.
I suggest replace the term "bound water capacity” with "maximum bound water capacity". The last term is more appropriate as FSP.
I recommend the paper need do the minor improvement as it was pointed out above.
Reviewer 2 Report
This is a very nice paper. It is about testing a hypothesis that explains why acetylated wood is more resistant to brown rot decay than untreated wood. If I understood it correctly, the hypothesis is that iron oxalate and acetate compounds associated to the fungus diffuse into the wood and degrade the fibre, which in turn becomes fungus’ food. According to the paper, it is believed that the diffusion of iron compounds into acetylated wood is much slower than into untreated wood, thus wood degradation is also much slower.
The authors measured the diffusion rate of iron compounds into wood and found that there are not diffusion rate differences between treated and untreated wood, which is contrary to the current understanding of the process. Even though the paper does not elucidate the actual mechanism of protection, it shows according to the authors that “much remains to be understood about the relationship between acetylation, diffusion, and brown rot decay”.
Since I am not an expert in brown rot decay, I cannot tell if there are better explanations that the authors are not aware of. However, if the assumption of the paper is correct, then this is a very interesting study that challenges current knowledge and speculate about possible explanations. The English is perfect, and the manuscript contains a very detailed description of all experiments and data analysis.
However, I would also recommend the Editor, and maybe the Authors, to consider adding a figure explaining the experiments. Even though everything is explained in the text, it is not very easy to follow by reading. It would have been much easier to understand if the paper had a figure showing the diffusion process: first the flow of solution into the lumens in the longitudinal direction, and then the diffusion of iron through the cell wall in the perpendicular direction. I had to read the manuscript two times to understand how diffusion occurred during these experiments.
Author Response
Thank you for the suggestion. We have added a figure (#1, line 120) to the text to show the relevant details.